# ETPTA Inverse Photonic Crystals for the Detection of Alcohols

**Matin Ashurov** [1,2,*] **, Stella Kutrovskaya** [1,2,3] **, Alexander Baranchikov** [4] **, Sergey Klimonsky** [5,*]
**and Alexey Kavokin** [1,2]

1   School of Science, Westlake University, 18 Shilongshan Road, Hangzhou 310024, China;
    stella.kutrovskaya@westlake.edu.cn (S.K.); a.kavokin@westlake.edu.cn (A.K.)
2   Institute of Natural Sciences, Westlake Institute for Advanced Study, 18 Shilongshan Road,
    Hangzhou 310024, China
3   Moscow Institute for Physics and Technology, Institutskiy Pass 9, Dolford 141701, Russia
4   Kurnakov Institute of General and Inorganic Chemistry of the Russian Academy of Sciences,
    Moscow 119991, Russia; a.baranchikov@yandex.ru
5   Faculty of Materials Science, Lomonosov Moscow State University, Moscow 119991, Russia
*   Correspondence: ashurov.m@westlake.edu.cn (M.A.); klimonskyso@my.msu.ru (S.K.)

**Abstract:** We developed a comparatively simple and inexpensive approach for the determination of the concentration of alcohols in water. The method is based on the study of the optical properties of ethoxylate trimethylolpropane triacrylate (ETPTA) inverse photonic crystals (IPhCs). The position of the transmission minimum associated with the first photonic stop band (PSB) is used as the analytical signal. The PSB position depends on the swelling degree of ETPTA photoresist and the refractive index of the tested alcohols and their mixtures with water. The signal increases linearly with increasing concentration of ethylene glycol and increases nonlinearly but monotonically with the concentration of methanol and ethanol in water. Sensitivity to alcohols, in the case of the ethylene glycol–water mixtures, reached about 0.55 nm/v.% or 560 nm/RIU (refractive index unit), which is sufficient for various applications in bio/chemical detection and environmental monitoring.

**Keywords:** photonic crystals; inverse opal; stop band; swelling; refractive index; sensors





## 1. Introduction

Alcohols are one of the most important organic solvents containing the hydroxy functional group (–OH) that is bonded to the carbon atom of an alkyl or substituted alkyl. They can be converted to and from many other types of compounds and are often used in the production of pharmaceutical preparations and organics. In addition, water–alcohol mixtures have been actively studied in connection with the internal richness of physical and chemical processes [1]. In the case of methanol–water solutions, the refractive index, the diffusion coefficient, or the viscosity show anomalies related to segregation and clustering of water and methanol molecules at about 70 mol.% water [2]. Ethanol–water mixtures show similar anomalies in the refractive index and viscosity at about 40 mol.% water and 75 mol.% water, respectively. Refractometry is one of the classical methods to estimate the water content in alcohols and offers physical hints about the molecular evolution in water–alcohol systems [1,3]. The nonmonotonic character of the water content dependence of the refractive index $n$ in methanol–water and ethanol–water mixtures has been studied for years. In addition to refractometry, there are other spectroscopic methods, such as UV-Vis spectroscopy and chromatography, but these methods have their own challenges and expenses.

Photonic crystals (PhCs) are a type of material with an ordered structure whose dielectric constant changes periodically. They are characterized by the presence of photonic band gaps in the electromagnetic wave spectrum [4,5]. Nowadays, there is a fairly wide range of applications of PhCs, with sensing being one of the most important ones [6,7]. Some PhC sensor applications include a signal enhancement resonator [8–11] and colorimetric

sensors that detect crystal parameter changes stemming from the external environment, such as temperature [12], refractive index [13], mechanical stimuli [14], electric field [15], pH/ionic strength [16,17], multi-tunable external stimuli [18], and chemical or biological agents [19–23]. A change in the spectral position of the photonic stop band (PSB) and, as a consequence, a change in the color of a PhC, in the case of its interaction with a gaseous or liquid chemical compound, can occur in two ways. The first is the refractometric method, in which the shift of the PSB occurs due to a change in the effective refractive index of the PhC as a result of the filling of structural voids with an analyte. The second approach, that we shall henceforth refer to as a "chemical" approach, is associated with the possibility of swelling [16,17,22,23] or compressing [18,24] of the PhC materials which leads to a change in the PhC lattice parameters. Both mechanisms can affect the spectral position of the PSB simultaneously [25–28]. Sensors based on PhCs are quite convenient and suitable for testing alcohols. Since the beginning of the 21st century, there has been rapid progress in the development of PhC sensors for the detection of alcohols. Asher's group developed a pH- and ethanol-sensitive material based on colloidal arrays incorporated with hydroxyethyl methacrylate hydrogel [29]. Pan et al. reported polyacrylamide inverse opal (IO) hydrogel to detect liquid alcohols [30]. Shi et al. studied the detection of methanol and ethanol by hydrogel inverse PhC [31]. However, the usability of hydrogel materials after long-term storage cannot be guaranteed. In addition, the use of special recognizing agents attached to the hydrogel leads to a very high sensitivity and, at the same time, very complex, nonlinear dependences of the analytical signal on the analyte concentration [16,29].

It is known that IO sensors possess higher porosity and sensitivity compared to opal sensors [32,33]. This comes from the fact that the IO is more porous; therefore, the analyte filling volume is threefold greater than that of an opal. On the other hand, inversion can lead to the destruction of the periodicity of the structure. It is especially dangerous if sintering is used in the inversion process. In this case, compression, cracking, and the appearance of various defects can be observed [34]. Therefore, there is a need for soft, nonannealing methods for obtaining IOs. These include inversion via hydrogel polymerization [10,29–31] or a photocurable resin [35–38] such as a PAMA in [35] (refractive index $n$ = 1.64), an NOA 61 in [36] ($n$ = 1.56), a negative photoresist (SU-8) in [37] ($n$ = 1.6), and trimethylolpropane ethoxylate triacrylate (ETPTA) in [38–40] ($n$ = 1.67 according to [38]). Successful polymerization of most hydrogels requires many hours of UV exposure or heat treatment, but for photopolymerization of resin, a few minutes is sufficient. An ETPTA photoresist compares favorably with both hydrogels and other photocurable resins due to its high refractive index. As was shown in [35,38], inversion with a photoresist completely retains the quality of the structure of the original template. At the same time, ETPTA IO films have not only a more porous structure than silica opal films, but also better mechanical strength and are characterized by a higher reflection peak in the range of the photonic stop band [38]. In our previous work, we explored the application of ETPTA IO PhC for alcohols sensing [39]. However, the investigation was limited to only two alcohols with one sample without a detailed study. These drawbacks are overcome in the present work, which presents a detailed study of the applicability of the ETPTA IO films for the detection of alcohols and their aqueous mixtures.

## 2. Experimental Section

### 2.1. Materials

Tetraethyl orthosilicate (TEOS) for the synthesis of $SiO_2$ spheres, trimethylolpropane ethoxylate triacrylate (ETPTA) as a photoresist, and 2-hydroxy-2-methyl-1-phenyl-1-propanone (HMPP) as a photoinitiator were purchased from Sigma-Aldrich. Hydrofluoric acid (HF in water, 37 v.%) and ammonia (($NH_3 \cdot H_2O$, 28 v.%) were obtained from Reachem. Menzel-Glaser glass coverslips of 24 × 24 × 0.4 mm, thoroughly washed in ethanol and acetone, were used as substrates for the deposition of $SiO_2$ spheres.

### 2.2. Preparation of the ETPTA IPhCs

Monodisperse $SiO_2$ spherical nanoparticles were synthesized through sol–gel technique using the recipe of the seeded growth described earlier [41]. Spheres of any desired diameter in the range from 240 to 310 nm were obtained by varying the time of the regrowth. Initial nuclei of ~30 nm diameter was synthesized by the Stöber method [42]. PhC films with opal structure were grown on the glass coverslips by the vertical deposition method [43]. Silica opal film was covered by a second glass substrate with spacers forming a gap between the two glass plates. ETPTA liquid monomer with approximately 2 vol. % HMPP photoinitiator filled the gap by capillary rise (sandwich method, [31,44]) as described previously [38,45]. Then, the sandwich construction was irradiated with UV for 10 min. The photopolymerized film was pulled out from the gap between the glasses, and the silica spheres were removed by etching in 8 vol.% HF.

### 2.3. Apparatus

The microstructure of PhC films was studied by a Carl Zeiss NVision 40 scanning electron microscope (SEM) at 0.5 kV acceleration voltage, 30 mm aperture, and 4 mm working distance. No conductive layer was applied to the sample surface before measurements. The full transmittance spectra (direct beam plus scattering light going through the sample) were recorded on a Lambda 950 spectrophotometer (PerkinElmer) equipped with a white sphere. A quartz cuvette with an impregnated PhC film was placed in a Lambda-35 spectrophotometer. The beam from the halogen lamp was focused on the surface of the sample in the form of a strip with 1 mm width and 10 mm length. The transmission spectra $T(\lambda)$ were recorded at normal light incidence on the sample surface in the wavelength range from 400 to 800 nm with a scanning step of 0.2 nm and a monochromator slit width of 0.5 nm. The minima of the dependences $T(\lambda)$ corresponding to the PSB were found using a cubic approximation. Taking into account all errors, the accuracy of determining the minima of $T(\lambda)$ was about $\pm 0.5$ nm. After measurements, the IO films were dried in air (~1 min). In the case of ethylene glycol, they were first washed with ethanol and then dried in air. The refractive index and density values for alcohols and their mixtures with water were obtained from the handbook [46].

## 3. Results and Discussion

### 3.1. Characterization of ETPTA IPhCs

The SEM images (Figure 1) show the periodicity and uniformity of the prepared ETPTA IO PhC films. Their parameters are provided in Table 1. The mean diameter of ETPTA spherical voids is 240, 260, 290, and 310 nm for samples S1–S4, respectively; this is in good agreement with the diameter of the silica spheres in the initial template. The different sizes of spherical voids resulted in the blue, green, yellow, and red colors of the inverse PhC films (see insets to Figure 2). The full transmittance spectra shown in Figure 2 (recorded using the white sphere) have a sharp single dip, which indicates the high crystalline quality of the ETPTA IO films. The corresponding transmittance minimum red shifts from 470 to 508, 572, and 607 nm as the diameter of the silica spheres increases from 240 to 260, 290, and 310 nm, respectively. The decrease in the transmittance spectra in the UV region is associated with the absorption of light by the ETPTA photoresist.

### 3.2. The Response of the ETPTA IPhCs to Alcohols

The ETPTA IO PhC film (sample S2) was immersed into water, methanol, ethanol, isopropanol, n-butanol, and ethylene glycerol, respectively. The original structural color (in a dry state) of the IPhC film was sky-blue, and it turned to red when immersed in alcohols, i.e., a redshift of the PSB was observed in the transmission spectrum (Figure 3a). Thus, according to the Bragg–Snell law, under normal incidence of light, the redshift of

the transmittance minimum caused by the change of the refractive index of solvents or the swelling of ETPTA framework can be written as:

$$\lambda_{min} \propto n_{eff}dS \tag{1}$$

where $d$ is a lattice constant and depends only on the particle size of the $SiO_2$ colloid template, but after impregnation to water–alcohol solutions, changes to:

$$d' = d*S \tag{2}$$

where $S$ is the equilibrium swelling degree of the ETPTA photoresist, and $n_{eff}$ is the effective refractive index of the material which can be calculated using $n_{void}$ equal to the refractive index of the corresponding liquid as:

$$n_{eff} = \sqrt{n_{ETPTA}^2 f_{ETPTA} + n_{void}^2(1 - f_{ETPTA})} \tag{3}$$

**Table 1.** Average diameter and standard deviations of the spherical voids, spectral position of the transmittance minimum, and the thickness of ETPTA IO films.

| Sample Number | Diameter of the Spherical Voids, $D$, nm [1] | Wavelength of Transmittance Minimum of the IPhCs, $\lambda_{min}$, nm [2] | The Thickness of the IPhC Films, $d$, µm [1,2] |
| --- | --- | --- | --- |
| S1 | 240 ± 8 | 470 | 5.60 ± 0.10 |
| S2 | 260 ± 7 | 508 | 3.72 ± 0.15 |
| S3 | 290 ± 8 | 572 | 1.95 ± 0.15 |
| S4 | 310 ± 9 | 607 | 3.03 ± 0.03 |

[1] Controlled using SEM. [2] Dates from transmittance spectra.

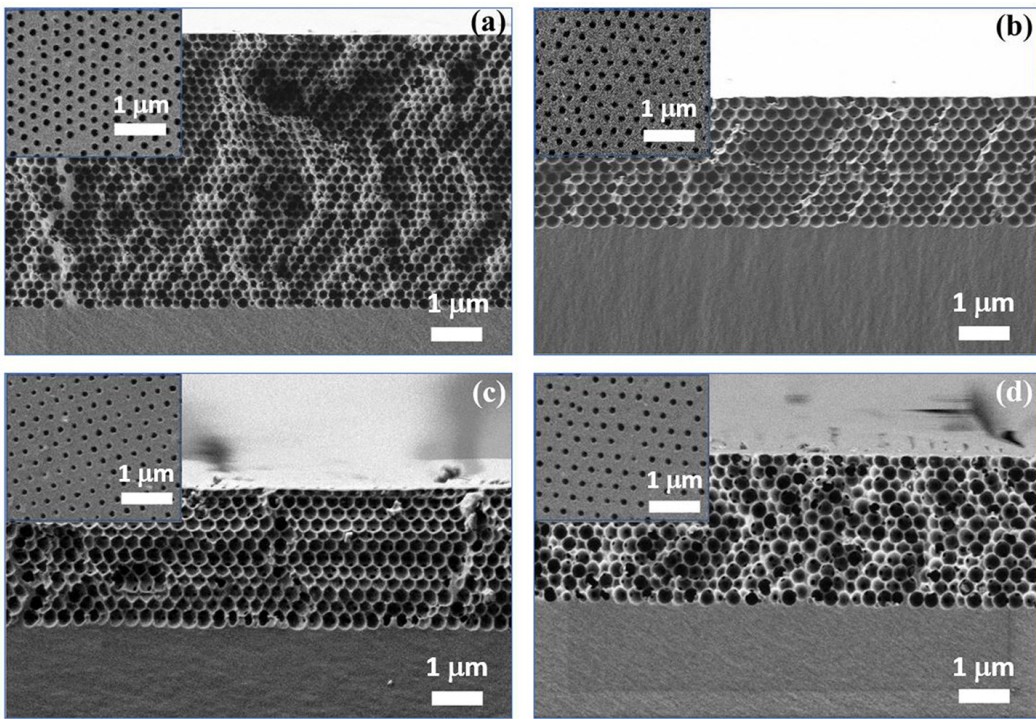

**Figure 1.** (**a**–**d**) SEM images of cross sections for all investigated ETPTA IO films: S1, S2, S3, and S4, respectively. Insets show magnified top-view SEM images of these films.

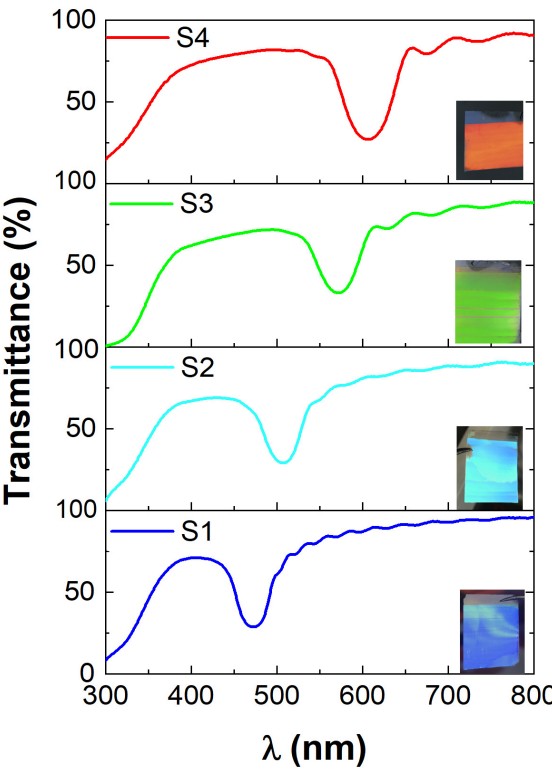

**Figure 2.** Transmittance spectra of the investigated ETPTA IO films recorded at the normal incident angle. Insets show optical micrographs of these samples.

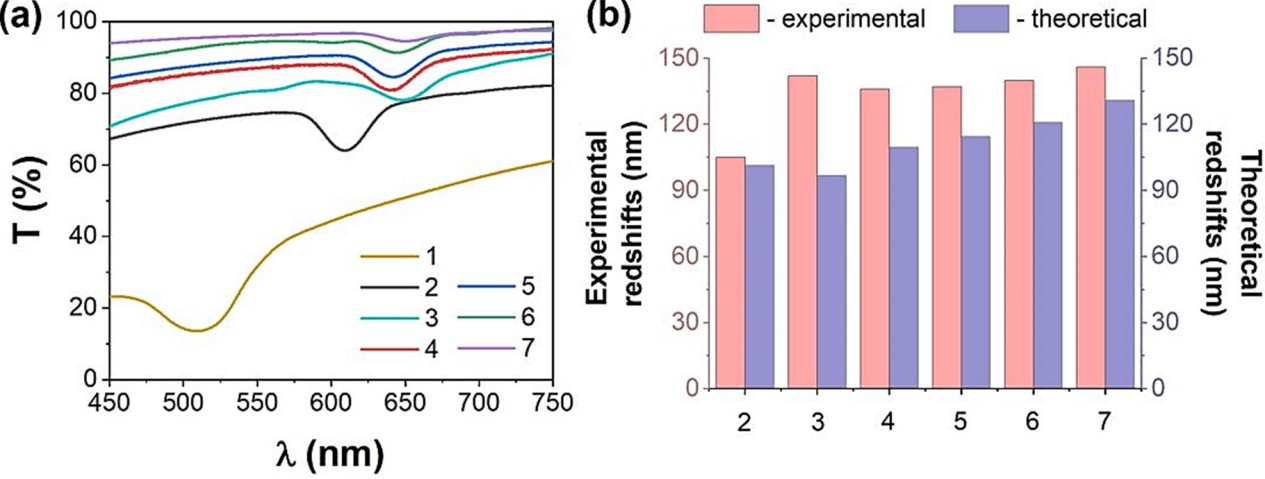

**Figure 3.** (**a**) Transmittance spectra $T(\lambda)$ and (**b**) corresponding bars of experimental and theoretical redshifts of the transmittance minimum for sample S2 immersed into various liquids: 1—air (dry state), 2—water, 3—methanol, 4—ethanol, 5—isopropanol, 6—n-butanol, and 7—ethylene glycol. The direct beam transmittance spectra shown in panel (**a**) were recorded using the Lambda-35 spectrophotometer.

In the above equation, $n_{void}$ is the refractive index of filling liquid, $n_{ETPTA}$ = 1.67 is the refractive index of ETPTA [38], and $f_{ETPTA}$ = 0.26 is the volume fraction of ETPTA in the IO structure. Figure 3b shows the experimental and theoretical redshifts of the transmittance minimum for the sample S2 in various liquids. The theoretical redshifts were calculated according to Equation (1) under the assumption that only the effective refractive index is changed. Under this assumption, the redshift value should change in the same order as the refractive index of the investigated liquids changes: 1.3280, 1.3330, 1.3614, 1.3775,

1.3993, and 1.4318 for methanol, water, ethanol, isopropanol, n-butanol, and ethylene glycol, respectively. However, as can be seen from Figure 3b, the experimental redshift values do not follow this sequence. For example, ethanol, isopropanol, and n-butanol have very close redshifts despite their different refractive index values. The reasonable explanation for this discrepancy is that the ETPTA matrix swells differently in different alcohols, and this swelling also shifts the wavelength of the stop band.

So, according to Equations (1) and (3), $S$ and $n_{void}$ are the factors that may affect $\lambda_{min}$. If a dry film of ETPTA IO is impregnated with water, the value of $n_{eff}$ will change. The refractive indices of water and ETPTA photoresist are 1.333 and 1.67, respectively. In the dry state, the pores of the IO film are filled with air, the refractive index of which is approximately 1; therefore, according to (3), the effective refractive index will be 1.21. When the ETPTA IO film is immersed in water, the value of $n_{eff}$ changes to 1.4283 (1.18 times 1.21). The $\lambda_{min}$ of the IO in water is 610 nm, which is 1.201 times that of the dry sample ($\lambda_{min}$ = 508 nm). There is little difference between the change in the effective refractive index and the change in $\lambda_{min}$ predicted by Equation (1). This is due to the swelling of ETPTA IO film in water. Even though the ETPTA IOs do not dissolve in water after polymerization, they can still swell in water. Using Equation (1) we can calculate the equilibrium swelling degree of the ETPTA 1.013 in water. As shown in Figure 4a–c, when the concentration of alcohols in the mixtures increases (from pure water to pure alcohol), the transmittance minimum redshifts by about 38 nm, 32 nm, and 41 nm for methanol, ethanol, and ethylene glycol, respectively. The overall redshift is due to the simultaneous increase of $S$ and $n_{eff}$. If more alcohol was present, the effective refractive index increased, and this caused a redshift in the transmittance spectra. Despite the fact that the refractive index of methanol–water and ethanol–water mixtures does not change monotonically, we still observed the monotonic concentration dependences for these mixtures (Figure 4d). This monotony is ensured by the swelling of the ETPTA photoresist in these mixtures. The effective refractive indices of the IO film immersed in the different mixtures were calculated using Equation (3), and the results are shown in Figure 4e. As we will see for methanol and ethanol mixtures with water, we observed an anomaly related with a nonmonotonic change in the effective refractive index due to the nonmonotonic changes in their refractive indices [46], whereas for ethylene glycol-water mixtures, it was almost linear.

Thus, ETPTA inverse opal swell the least in ethylene glycol and the most in methanol; that is, the value of $S$ was the highest in methanol and the lowest in ethylene glycol (Figure 4f). The contributions of various alcohols to $n_{eff}$ and $S$ are different but similar overall, so the observed redshifts (i.e., changes in $\lambda_{min}$) are close to each other. In general, our ETPTA photoresist swells. The equilibrium volume of the nonionic ETPTA is determined by the condition that the total polymer osmotic pressure is [47]:

$$P_T = P_M + P_E = 0 \tag{4}$$

where $P_M$ is the osmotic pressures due to the free energy of mixing, and $P_E$ is the polymer network elastic restoring force. The polymer osmotic pressure which arises from the free energy of mixing of the polymer with solvent is [16]:

$$P_M = -\frac{RT}{V_s}\left[\ln\left(1 - \frac{V_0}{V}\right) + \frac{V_0}{V} + \chi\left(\frac{V_0}{V}\right)^2\right] \tag{5}$$

where $R$ is the universal gas constant, $V_0$ is the dry polymer volume, and $V_s$ is the molar volume of the solvent (located into 26 vol.% of the IO film). The osmotic pressure which results from the network elasticity for uniform 3D swelling is:

$$P_E = -\frac{RT n_{cr}}{V_m}\left[\frac{1}{2}\frac{V_0}{V} - \left(\frac{V_0}{V}\right)^{1/3}\right] \tag{6}$$

where $n_{cr}/V_m$ is now the density of cross-linked chains in the network measured in molarity. We initially calculate this density, 0.45 mol/L, from the crosslinker to ETPTA stoichiometry. Also, we took advantage of the fact that $\varphi = (V_0/V) = (\lambda_0/\lambda)^3$, where the $\lambda_0$ and $\lambda$ are the diffracted wavelengths for dry and swollen ETPTA, respectively. We can use Equations (4)–(6) to calculate $\chi$ (the Flory–Huggins parameter for various alcohol–water solutions (Figure 5)).

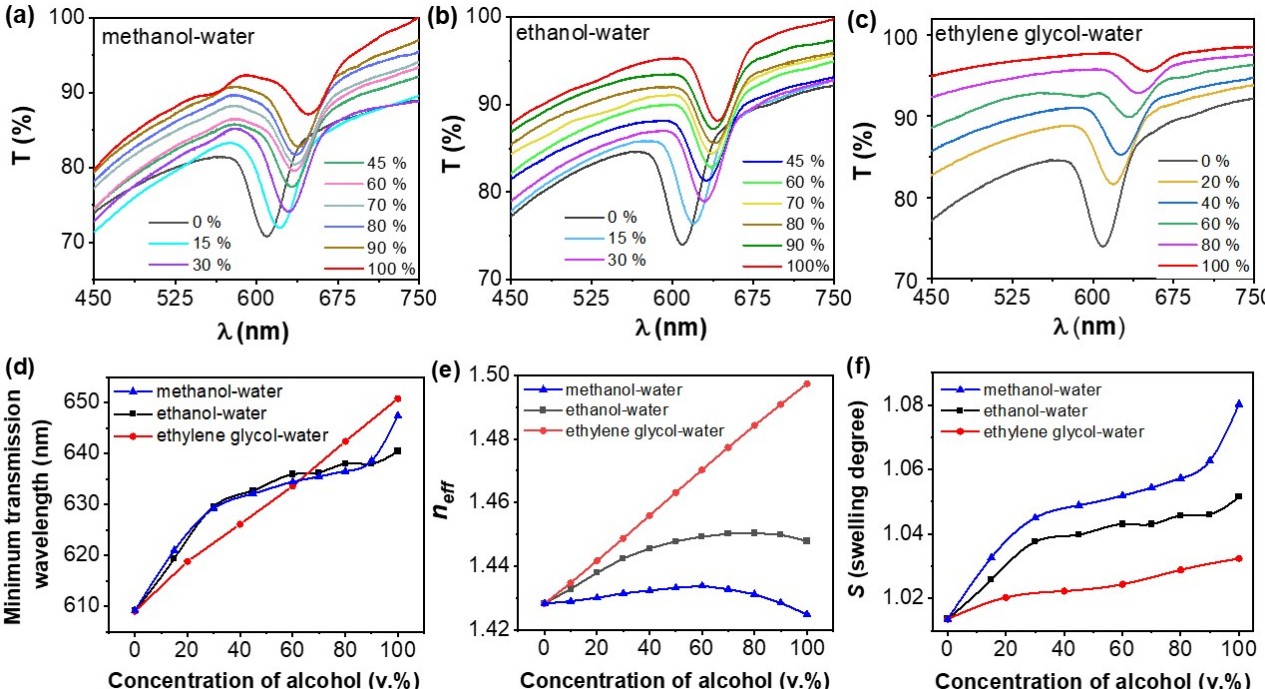

**Figure 4.** (**a**–**c**) Transmittance spectra of the ETPTA IO film (sample S2) soaked into different ratio of the alcohol−water mixtures: methanol, ethanol, and ethylene glycol, respectively; (**d**) minimum transmission wavelength, (**e**) calculated effective refractive indices, and (**f**) equilibrium swelling degree of the IO film vs. concentration of alcohols in water. All percentages of alcohol concentration are provided in volume%.

The calculated value of $\chi$ for ETPTA in pure water is 3.248 and in methanol, ethanol, and ethylene glycol, it is 1.219, 1.502, and 1.875, respectively. This means that methanol interacts stronger with the ETPTA photoresist than ethanol and water. A lower $\chi$ implies higher ETPTA–solvent interaction and thus indicates a higher solubility. We also observed a monotonic dependence of $\chi$ on the concentration of these alcohols in water.

Spectral redshifts deduced from impregnation of ETPTA IOs with different void diameters to ethylene glycol–water solutions are summarized in Figure 6a. The figure clearly illustrates the linearity of $\lambda_{min}$ *(c)* dependencies. The slope of the linear dependence $\lambda_{min}$ *(c)* for the ETPTA IO samples is steeper than for the SiO$_2$ opal films (linear dependence $\lambda_{min}$ *(c)* for the SiO$_2$ opals is shown in Figure S1).

However, in both cases, the experimental data does not agree with the Bragg–Snell law without taking into account the swelling of the PhC films. This is not a disadvantage of our sensors, since we rely on empirical calibrations of the dependence $\lambda_{min}$ *(c)* and its linearity. As we discussed above, the discrepancy is associated with swelling of ETPTA IOs. Also, from Figure 6b we can see that the swelling degree depends on the size of spherical voids in IO films, which was expected: if the pore size is larger, then the swelling is stronger. In Figure 6a, we can observe a slight increase in the slope of the linear dependence which characterizes the sensitivity of the sensors ($\Delta\lambda/\Delta c$ or $\Delta\lambda/\Delta n$): with an increase of the pore size, the period of IO PhCs and the degree of swelling, respectively, increase, which ultimately leads to a higher sensitivity value, from 0.4 nm/v.% up to 0.55 nm/v.%

(Figure 6c). Since the accuracy of determining the minimum $T(\lambda)$ using the cubic function is approximately 0.5 nm, this allows us to determine the concentration of ethylene glycol in water with an accuracy of about 1 vol.%.

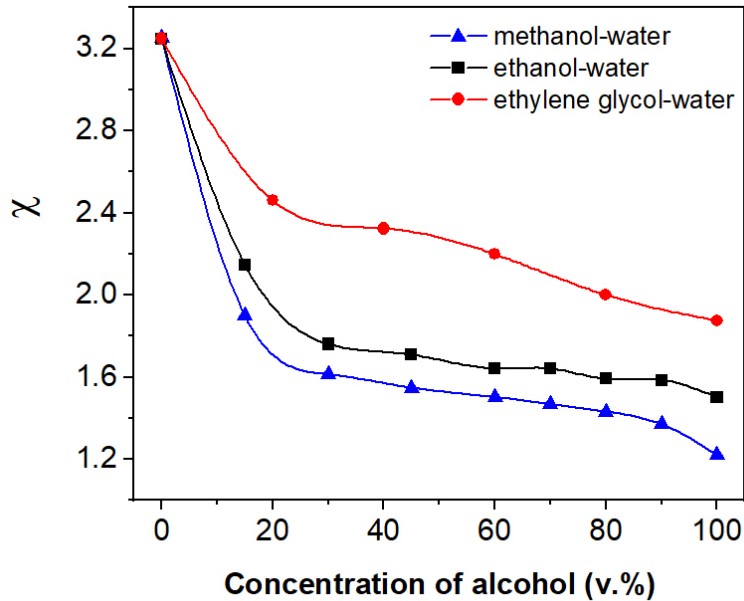

**Figure 5.** Flory–Huggins parameter $\chi$ for ETPTA IO film vs. concentration of alcohol in water.

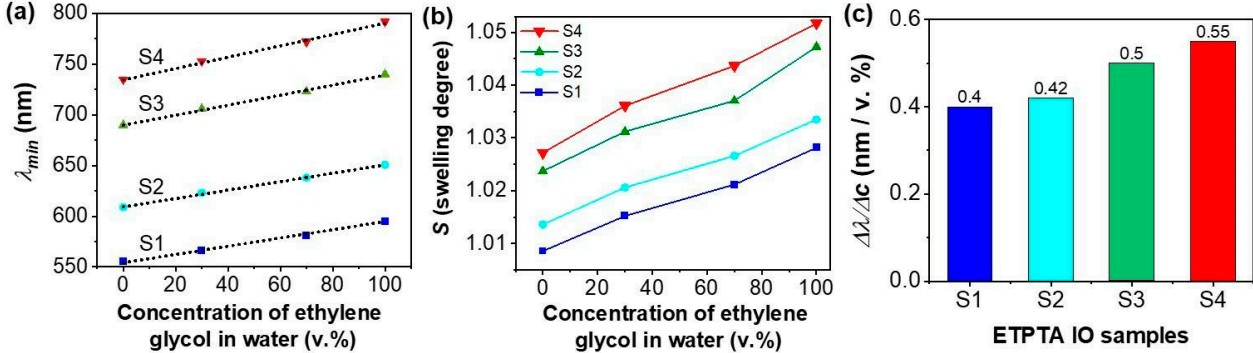

**Figure 6.** (**a**,**b**) Measured wavelength of transmission minimum and swelling degree of the ETPTA IO films with different spherical void diameters versus concentration of ethylene glycol in water; the measured data are represented by symbols; the dashed lines are linear fits to the experimental dependencies; (**c**) sensitivity of the investigated samples to ethylene glycol–water solutions per unit in changes of concentration.

The obtained value of the sensitivity of our sensor of 0.55 nm/vol.% can be easily converted to a change in the refractive index unit (560 nm/RIU) and compared with the sensitivity of pure refractometric sensors. According to the Bragg–Snell law, sensitivity to the refractive index is proportional to the interplanar distance $d$ in the crystal [33]. Therefore, it is necessary to compare sensors with close values of the periods of the structure. The Zheng group achieved sensitivity to ethanol (up to 70 vol.%) about 0.15 nm/vol.% using $TiO_2$ IO film with void diameter $D = 210$ nm [48]. In Ref. [49], the theoretical sensitivity of a sensor based on IO from $WO_3$ films with a structure period of 325 nm, which corresponds to a void diameter $D = 230$ nm, was about 270 nm/RIU, and its experimental sensitivity value was about 327 nm/RIU. Another example is refractometric sensors based on nanoporous anodic alumina [13,50–53]. These one-dimensional structures are widely used as sensors for detecting alcohols in water. Among our known publications, the highest sensitivity was reported in Ref. [53]: $\Delta\lambda/\Delta n = 441$ nm/RIU, which is less than the sensitivity reported in

the present work (560 nm/RIU). Law et al. presented Bragg reflectors based on nanoporous anodic alumina and characterized by the sensitivity of 390 nm/RIU [54]. The data of other publications collected in [13] vary from 71 to 164 nm/RIU. These comparisons lead us to conclude that the sensor material developed by us demonstrates a higher sensitivity compared to other PhC sensors based on purely refractometric measurement methods. In terms of sensitivity to low concentrations of ethanol in water (up to 40 v.%), our sensors can compete with previously proposed chemical sensors based on polyacrylamide [31], and their sensor sensitivity significantly exceeds the sensitivity of PhC sensors based on polydimethylsiloxane in the range from 0 to 90 wt.% ethanol in water [55]. It should also be noted that the chemical sensors proposed in Refs. [29,30] have a higher sensitivity to methanol and ethanol. However, in these works, complex procedures for preparation of sensors were used, including the introduction of a special sensitive agent into the PhC structure. In addition, a significant problem for most PhC sensors with the chemical response mechanism is the nonmonotonic and often poorly predictable character of the dependence of the position of the stop band on the concentration of alcohols in water. In particular, a complex nonlinear dependence of the response to ethylene glycol with an average sensitivity of about 0.25 nm/wt.% was achieved in [30]. The authors of [29] claimed that their sensors recognize ethanol concentration with an accuracy of 1% within the range up to 40 v.% ethanol in water, and the authors [30] positioned their sensors predominantly for qualitative colorimetric testing rather than quantitative control. In contrast to them, we have shown a linear dependence of the analytical signal on the concentration of ethylene glycol in water, which is convenient for practical control of its concentration of about 1%. We can also assume that such linear dependences can also be obtained for other water–alcohol solutions in which their refractive index changes linearly with composition. Let us also point out that refractometric sensors based on anodic alumina for the composition of liquids suffer from significant instability [13].

### 3.3. Recyclability of ETPTA IO Films

ETPTA IO films have good physical stability and are chemically inert to alcohols. The IPhCs were soaked into alcohol–water solutions and easily recovered from those liquids. After removing the PhC from the cuvette with the solution, the film was washed with water, dried in air, and reused. Therefore, it was important for us to know their response stability during repeated use. Figure 7 shows the results of 10 times impregnation with 40 vol.% ethanol followed by air drying at room temperature. In addition, we noticed that after all experiments that took over three months, the transmittance spectra did not change much (see Figure S2).

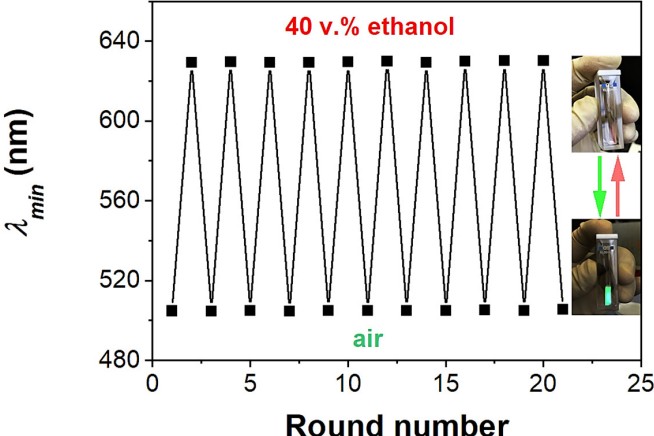

**Figure 7.** Recyclability of ETPTA IO film (tested in the dry state and in the wet state in 40 v.% ethanol–water solution). Inset shows photo of IO film in a quartz cuvette in both states.

## 4. Conclusions

ETPTA IO PhCs with spherical void sizes ranging from 240 to 310 nm were synthesized by a colloidal crystal templating method. The IO PhCs have PSB in the visible-light region and demonstrate bright structural colors. They are subject to fast and reversible changes in their structural colors and in the corresponding transmittance minimum in response to alcohols and their mixtures with water. These responses are mainly attributed to changes in the degree of swelling and the refractive index of the tested liquid filling the structural voids of the IO. Our results for water–alcohol mixtures show that the wavelength of the transmittance minimum linearly and monotonically increases with the concentration of ethylene glycol and increases nonlinearly with the concentration of methanol and ethanol. Our sensors are characterized by a sensitivity to the concentration of ethylene glycol of about 1%. A high refractive index sensitivity $\Delta\lambda/\Delta n = 560$ nm/RIU has been achieved. The ETPTA IO films are stable in alcohols, which is advantageous for various bio/chemical applications.

**Supplementary Materials:** The following supporting information can be downloaded at: https: //www.mdpi.com/article/10.3390/condmat8030068/s1, Figure S1: Sensitivity of the $SiO_2$ opal PhC films to ethylene glycol–water solutions; Figure S2: Transmittance spectrum of ETPTA IPhC film (sample S2) before and after all experiments (after 3 months) on impregnation with water–alcohol solutions and drying in air.

**Author Contributions:** Conceptualization, M.A. and S.K. (Sergey Klimonsky); formal analysis and investigation, A.B. and S.K. (Stella Kutrovskaya); writing—original draft preparation, M.A. and S.K. (Sergey Klimonsky); writing—review and editing, M.A., A.K. and S.K. (Sergey Klimonsky); supervision, A.K. and S.K. (Sergey Klimonsky); project supervision, A.K. All authors have read and agreed to the published version of the manuscript.

**Funding:** This research was funded by start-up funding for International Center for Polaritonics, Westlake University, Project 1012516021801.

**Data Availability Statement:** The data supporting the findings of this study are available from the corresponding author (M.A.) upon request.

**Acknowledgments:** The authors are grateful to colleagues who participated in experiments, especially Sh. Umedov and S. Ikrami (Faculty of Materials Science, Lomonosov Moscow State University). SEM images were obtained at the IGIC RAS Joint Research Center for Physical Methods of Research.

**Conflicts of Interest:** The authors declare no conflict of interest.

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
