# Peer review of "ETPTA Inverse Photonic Crystals for the Detection of Alcohols"

_condensedmatter, doi:10.3390/condmat8030068_

Round 1
Reviewer 1 Report
1.
1. In Figure 1, the authors removed the scale from the SEM images. In my opinion, returning it would make the figures more informative.
2. If we consider in detail the dimension of sensitivity according to the formula ( ), it is possible to notice, that sensitivity really has dimension nm. That means that the sensitivity directly depends on the working wavelength, as for increase of sensitivity it will be enough simply to move to a long-wave part of a spectrum. In particular, in the article [Efimov I.M., et al. Physica Scripta, 97, 055506 (2022): https://doi.org/10.1088/1402-4896/ac5ff7] was raised the problem of a measuring this value. In this article it was proposed the following variant of sensitivity calculation by following formula ( ).
3. It would be interesting for readers to know how much time is it necessary to dry the PC to achieve repeatability between different iterations.
Author Response
Comment 1: In Figure 1, the authors removed the scale from the SEM images. In my opinion, returning it would make the figures more informative.
Answer: In Figure 1 the scale from SEM images have been returned.
Comment 2: If we consider in detail the according to the formula ( ), it is possible to notice, that sensitivity really has dimension nm. That means that the sensitivity directly depends on the working wavelength, as for increase of sensitivity it will be enough simply to move to a long-wave part of a spectrum. In particular, in the article [Efimov I.M., et al. Physica Scripta, 97, 055506 (2022): https://doi.org/10.1088/1402-4896/ac5ff7] was raised the problem of a measuring this value. In this article it was proposed the following variant of sensitivity calculation by following formula ( ).
Answer: Thank you for your deep interest about the sensitivity of our sensors. Yes, we agree with you that to obtain highly sensitive PhC sensors, we just need to move to the IR region, but sensors working in the visible region of the spectrum always have an advantage in terms of visualization and demonstration.
Thank you for the proposed publication on the sensitivity calculation. Really very interesting work, but using their formula in our case probably we will get approximately the same sensitivities as we have currently.
Comment 3: It would be interesting for readers to know how much time is it necessary to dry the PhC to achieve repeatability between different iterations.
Answer: After working with water-alcohol solutions (especially with methanol and ethanol), our PhC film dries instantly due to its low thickness. We waited about 1 minute before the next measurement. This time was added in the Experimental Part (Section 2.3.).

Reviewer 2 Report
Authors have presented some interesting experimental results those can be accepted. However, being a refractive index sensor authors have not commented on selectivity of the devices. This is a major concern in these types of devices. Authors needs to discuss a detailed selectivity prospective of proposed devices for methanol, ethanol, and ethylene glycol concentrations in water.
Additionally, as authors are using silica nanoparticles then why the transmission is very less?
Not Applicable
Author Response
Comment 1: Authors have presented some interesting experimental results those can be accepted. However, being a refractive index sensor authors have not commented on selectivity of the devices. This is a major concern in these types of devices. Authors needs to discuss a detailed selectivity prospective of proposed devices for methanol, ethanol, and ethylene glycol concentrations in water.
Answer: Thank you for your interest in the selectivity of our sensors. Our sensors are not purely refractometric as you mentioned, since to takes place the swelling of ETPTA photoresist. Actually, the swelling of our photoresist helps to some extent to the selectivity of our sensors, but this issue was not considered in details in this work. Our work is aimed more at the quantitative control of the concentration of a known alcohol than at the identification of various alcohols.
Comment 2: Additionally, as authors are using silica nanoparticles then why the transmission is very less?
Answer: The transmission spectra that are presented in our work are not from an opal silica film, but from inverse opal film based on ETPTA photoresist, which is transparent in the visible region of the spectrum. The low transmission value may be due to the scattering on imperfections of the inverse opal structure. This factor can be minimized when working with a white sphere that collects scattered light. For example, in Fig. 2, where the spectrum was recorded using a Lambda 950 spectrophotometer equipped with a white sphere, the transmission is quite high.

Reviewer 3 Report
Review comments file attached.

Author Response
Comment 1: The authors claim that the proposed ETPTA inverse opal is advantageous over the other studies for different alcohol sensing. In the reported results the ETPTA opal spectral response (Fig. 4d) is almost the same for both methanol-water and ethanol-water mixtures from 0 to 90 v.% concentration. So, the authors claim of sensing these alcohols is not valid without a noticeable wavelength difference. Moreover, at 100 % alcohol concentration the spectral wavelength difference between methanol, ethanol and ethylene glycol is about only a +/- 5nm. These values could be within the error estimation for the identification of center of photonic bandgap. Authors should address this concern in detail.
Answer: Yes, indeed, for mixtures of water-ethanol and water-methanol up to 90 v. % concentration of alcohol, the curves look very closely. The issue of selectivity, which you mentioned, is not in the aim of this work. The main advantage of our sensors over purely refractometric sensors is an absence of problem of ambiguity of the obtained data due to the fact that the dependences of the response of our sensors on the concentration of any alcohol are monotonic. As for the accuracy of determining the minimum transmission, it is about 0.5 nm (details of the experimental errors are described in the experimental section), so all three alcohols can really be distinguished at 100 % concentration. Note that the size of the symbols for the dependences of the wavelength vs composition of liquids is approximately equal to the accuracy of their determination.
Comment 2: In Fig. 7, the recyclability of the opal was reported for 0 to 40% range. Does it work for other concentrations and other opals used in this study?
Answer:
We see no reason why the recyclability should be worse at other concentrations, and we are sure that the same recyclability will be observed for other samples.
Comment 3: The S2 inverse opal transmission spectrum reported in Fig. 2 and Fig.3a does not show the same percentage of PBG depth.
Answer: Yes, indeed, because they were taken with different spectrophotometers (Lambda 950 and Lambda 35). In the case of Lambda 950 spectrophotometer we recorded the full transmittance (the direct and scattered beams were summed with the help of the white sphere), whereas in the case of Lambda 35 only the direct beam was registered. This feature of our work is specified in the experimental section. Nevertheless, the following additions have been inserted into the text for clarity.
Page 3, Section 3.1: “The transmittance spectra shown in Fig. 2 have a sharp single dip” has been replaced with “The full transmittance spectra shown in Fig. 2 (were recorded using the white sphere) have a sharp single dip”.
The caption to Fig. 3: the sentence “The direct beam transmittance spectra shown in panel (a) were recorded using the Lambda-35 spectrophotometer” has been added.
Comment 4: In Fig. 4, irrespective of alcohol type the increasing alcohol concentrations is leading to reduced PBG contrast (< 5%) and becoming almost transparent. So, this reduced contrast could not certify the potential of the opals for alcohols sensing.
Answer: Yes, indeed, as the refractive index of the analyte approaches to the refractive index of the material, the contrast gradually disappears. This problem exists in all polymeric PhC sensors. The advantage of our sensor is that we took a photoresist with a high refractive index (about 1.7). The corresponding optical contrast is sufficient for reliable control of the PBG when working with all the studied liquids.
Comment 5: The figures are poor quality and need to be improved. The authors should also expand the ETPTA in the title.
Answer: The quality of the figures has been improved as far as possible.
It seems to us that when expanding the abbreviation ETPTA, the title will turn out to be too long, so we left it unchanged. We believe that the decoding of the ETPTA in the abstract is sufficient for a correct understanding.
Comment 6: Authors missed citing some of the earlier reports on similar topics and the application of opals for other kind of sensing applications.
Answer: Mentioned references have been added to the list of references (numbers 8, 28 and 32).
